# MEASURING FEATURE SPARSITY IN LANGUAGE MODELS

## ABSTRACT

Recent works have proposed that intermediate activations in language models can be modelled as sparse linear combinations of vectors corresponding to features of the input text. Under this assumption, these works have aimed to reconstruct these feature directions using sparse coding. We develop metrics which can be used to assess the success of these sparse coding techniques and thereby implicitly test the validity of the linearity and sparsity assumptions. We show that our metrics can predict the level of sparsity on synthetic sparse linear activations, and that they can distinguish between sparse linear data and several other distributions. We use our metrics to measure the level of sparsity in several language models. We find evidence that language model activations can be accurately modelled by sparse linear combinations of features, significantly more so than control datasets. We also show that model activations appear to be sparsest in the first and final layers, and least sparse in middle layers.

## 1 INTRODUCTION

Over the past decade, neural networks have demonstrated remarkable performance in many domains, including natural language processing (Brown et al., 2020; OpenAI, 2023), computer vision (Wang et al., 2022), protein structure modelling (Jumper et al., 2021) and complex strategy games (Silver et al., 2017; Berner et al., 2019). However, our capacity to interpret these models lags far behind. Being able to reliably and automatically determine the features used by such models – that is, the properties of the model inputs that the model extracts and uses for its predictions – would be a major step forward in our ability to interpret and safely deploy such models.

Many methods have been proposed for extracting model features, including feature visualization (Erhan et al., 2009; Olah et al., 2017), saliency maps (Simonyan et al., 2014) and feature importance scores (Lundberg et al., 2017). Feature extraction for language models is less well developed, though recent techniques can locate facts within models (Meng et al., 2022) and show how models perform certain linguistic operations (Geiger et al., 2021). Nevertheless, most approaches to language model interpretability require human input at key stages and scale poorly to extracting many features.

One particular obstacle is the existence of *polysemantic neurons* in language model MLP layers – neurons that respond to multiple seemingly unrelated features (Olah et al., 2020; Elhage et al., 2022). One hypothesised explanation for this behaviour, is that in the absence of co-occuring features, models use one neuron to represent several features without loss of accuracy. Toy models for this phenomenon, known as superposition, have been proposed by Elhage et al. (2022) and Scherlis et al. (2022). These models make two key assumptions: the *linear representation hypothesis*, which states that intermediate activation vectors in neural networks can be decomposed as a linear combination of vectors corresponding to individual features, and the *sparsity hypothesis*, which states that only a small number of features are active in any given input (Elhage et al., 2022).

Under these assumptions, decomposing language model activations into components corresponding to different features can be viewed as an instance of the sparse coding problem (Olshausen & Field, 1997; Elhage et al., 2022; Sharkey et al., 2022). Recent works have applied sparse coding techniques to language models and succeeded in automatically decomposing them into sparse linear combinations of human-interpretable feature vectors (Yun et al., 2021; Cunningham et al., 2023). However, these methods have typically assessed their decompositions by how easy it is to find plausible or

predictive descriptions of the set of inputs on which a particular found feature is active. Such metrics can indicate how successful these methods are at finding interpretable feature directions, but say little about the accuracy of the underlying assumptions of linearity and sparsity.

The main contribution of this work is to introduce more rigorous and quantitative ways of measuring the success of sparse coding methods applied to language models. This allows us to properly assess the extent to which activations can accurately be modelled as sparse linear combinations of feature vectors, and see how the level of sparsity depends on properties of the model. In summary, we:

- Propose novel metrics for measuring the success of sparse coding on neural network activations, and demonstrate their robustness on synthetic activations;

- Use our metrics to provide quantitative evidence that neural network activations can be accurately modelled as sparse linear combinations of feature vectors, supporting claims in earlier works (Yun et al., 2021; Cunningham et al., 2023);

- Provide a more thorough analysis of the success of sparse coding as compared to previous works, including studying the relative sparsity of various model types and across layers.

## 2 BACKGROUND

Formally, the *linear representation hypothesis* can be interpreted in the following way. At a fixed layer of a given neural network, we assume that the activations represent $m$ different ground-truth features $\mathcal{F}_1, \ldots, \mathcal{F}_m$, each encoded by a feature vector $\mathbf{f}_1, \ldots, \mathbf{f}_m \in \mathbb{R}^d$. An input containing features $\mathcal{F}_{i_1}, \ldots, \mathcal{F}_{i_k}$ should be represented by an activation vector which is a linear combination of $\mathbf{f}_{i_1}, \ldots, \mathbf{f}_{i_k}$. Conversely, any activation $\mathbf{x} \in \mathbb{R}^d$ should be approximately decomposable as a linear sum $\mathbf{x} \approx \alpha_{i_1} \mathbf{f}_{i_1} + \cdots + \alpha_{i_k} \mathbf{f}_{i_k}$ where $\mathcal{F}_{i_1}, \ldots, \mathcal{F}_{i_k}$ are the features present in the input producing activation $\mathbf{x}$ and $\alpha_{i_1}, \ldots, \alpha_{i_k}$ are non-negative feature coefficients. We call $m$ the *dictionary size* and $d$ the *embedding size*. For notational convenience we combine the feature vectors into a matrix $\Phi \in \mathbb{R}^{d \times m}$ whose columns are $\mathbf{f}_1, \ldots, \mathbf{f}_m$. The linear representation hypothesis was formulated by Arora et al. (2018) and Elhage et al. (2022), motivated by observations of apparent linear structure in the representations learned by generative models (Bojanowski et al., 2018; Burns et al., 2023; Ilharco et al., 2022; Turner et al., 2023).

The *sparsity hypothesis* says that a typical activation $\mathbf{x}$ only requires a small number of features to approximately represent it as a linear combination of those features, that is, we can find $\alpha_{i_1}, \ldots, \alpha_{i_k}$ such that $\mathbf{x} \approx \alpha_{i_1} \mathbf{f}_{i_1} + \cdots + \alpha_{i_k} \mathbf{f}_{i_k}$ and $k \ll d$ (Arora et al., 2018; Elhage et al., 2022). In practice, we expect that most human-interpretable features of an input should be active on only a small number of inputs, making sparsity a natural assumption (Arora et al., 2018; Elhage et al., 2022).

In our language model setting, we observe a set of intermediate activations $\mathbf{x}^{(1)}, \ldots, \mathbf{x}^{(n)}$ from a given layer. Our task is to reconstruct the feature vectors $\mathbf{f}_1, \ldots, \mathbf{f}_m$ corresponding to the ground-truth features. If the set of activations is fixed, we may combine them into a matrix $X \in \mathbb{R}^{d \times n}$ with columns $\mathbf{x}^{(1)}, \ldots, \mathbf{x}^{(n)}$ and formulate our problem as finding $\Phi \in \mathbb{R}^{d \times m}$ and $\alpha \in \mathbb{R}^{m \times n}$ such that $X = \Phi\alpha + \varepsilon$ where $\alpha \succeq 0$ is sparse and $\varepsilon$ is small (typically in $L^2$ norm).

In practice, since we can generate intermediate activations at will by running our model on new text, we can alternatively imagine that we have an activation distribution from which we can sample. Then, our problem amounts to finding $\Phi \in \mathbb{R}^{d \times m}$ such that for an activation $\mathbf{x}$ drawn from this distribution we can find $\alpha \in \mathbb{R}^m$ and $\varepsilon$ such that $\mathbf{x} = \Phi\alpha + \varepsilon$, where $\alpha$ is typically sparse and $\varepsilon$ is small in expectation. For convenience, in this paper we will stick to notating the case of a fixed matrix $X$ of activations, but all quantities discussed can be trivially extended to the case where we draw samples $\mathbf{x}$ from the distribution of activations – wherever we sum over rows of $X$ or $\alpha$, this should instead be replaced by an expectation over the activation distribution.

For a fixed matrix $X$ of activations, the sparse coding problem is frequently solved by minimizing the *sparse coding* objective

$$\mathcal{L}(\Phi; \alpha) = \frac{1}{n} \left( \|X - \Phi\alpha\|_2^2 + \lambda \|\alpha\|_1 \right) \tag{1}$$

over $\alpha \in \mathbb{R}^{m \times n}$ and $\Phi \in \mathbb{R}^{d \times m}$, where the $L^1$ norm can be viewed as a continuous relaxation of the $L^0$ sparsity norm and $\lambda$ is some hyperparameter controlling the degree of sparsity regularization. In

this work, we constrain the columns of $\Phi$ to have $L^2$ norm 1 and minimise $\mathcal{L}(\Phi)$ using an iterative optimization procedure similar to that of (Beck & Teboulle, 2009; Yun et al., 2021), as explained in Appendix A.

For our metrics defined later, it will be convenient to define $\alpha(\Phi) = \arg\min_{\alpha \in \mathbb{R}^{m \times n}} \mathcal{L}(\Phi; \alpha)$, i.e. the optimal choice of $\alpha$ for a given $\Phi$, and $\mathcal{L}(\Phi) = \mathcal{L}(\Phi, \alpha(\Phi))$, i.e. the value of the sparse coding objective for a given $\Phi$, assuming an optimal choice of $\alpha$. In practice, we approximate $\alpha(\Phi)$ using the same optimization procedure described in Appendix A for minimizing (1).

## 3    METRICS FOR SUCCESS OF SPARSE DICTIONARY LEARNING

For a sparsity metric to be meaningful, it should be invariant under scaling all activations by the same factor, continuous with respect to the activations (since we expect the activations not to be an exact linear combination of the feature vectors, so our metric should be robust to small perturbations) and ideally intuitive. In particular, for data which is genuinely generated using a sparse linear mechanism with an average of $k$ fully active features, we'd like our metric to be approximately $k$.

We consider four classes of metrics, the first two of which have been previously considered (Sharkey et al., 2022) and the second two of which are novel as far as we are aware.

**Non-zero entries:** The most natural metric of success is the average number of non-zero entries in the coefficient vector $\alpha$, which we denote $\mathcal{N}_0(\Phi) = \frac{1}{n}\|\alpha(\Phi)\|_0$. Though $\mathcal{N}_0(\Phi)$ is intuitive and invariant under scaling, we find that it is not robust in practice.

**Final loss value:** Second, we consider using $\mathcal{L}(\Phi)$ for the final value of $\Phi$, as done by Sharkey et al. (2022). Though this metric is continuous, it is not invariant under scaling and so does not give useful comparisons across different models or layers within a model.

**Average coefficient norm:** Third, we consider $\mathcal{S}_p(\Phi) = \|\alpha(\Phi)\|_p^p / \|\alpha(\Phi)\|_\infty^p$ for each $p > 0$. This corresponds to the average $L^p$ norm of the coefficient vector, normalized by the average maximum coefficient. This is scale invariant and, in the case where all feature coefficients are either zero equal to the same positive value, $\mathcal{S}_p(\Phi)$ corresponds to the average number of features present. In practice, we typically use $p = 1$ for simplicity and robustness.

**Normalized loss:** We define the normalized loss as $\mathcal{L}_{\text{norm}}(\Phi) = \mathcal{L}(\Phi)/(\lambda\|\alpha(\Phi)\|_\infty)$. This is similar to the final loss value, but normalized by the average magnitude of the feature coefficients, restoring invariance to scaling. Intuitively, if we perfectly reconstruct the activations so that $\mathbf{x} = \Phi\alpha$ always, then $\mathcal{L}(\Phi) = \lambda\|\alpha(\Phi)\|_1$ and so $\mathcal{L}_{\text{norm}}(\Phi) = \mathcal{S}_1(\Phi)$. In practice, reconstruction is somewhat imperfect, in which case $\mathcal{L}_{\text{norm}}$ also includes a penalty for the unreconstructed part. This makes it more stable over a range of hyperparameters.

### 3.1    EXPERIMENTAL VERIFICATION OF METRICS

We now test the effectiveness of the four proposed metrics by evaluating them on various synthetic datasets where we know the true level of sparsity. First, we test how well our metrics can predict the true level of sparsity for sparse linear data. To do this, we generate synthetic activation distributions that satisfy the sparsity and linearity hypotheses with varying average numbers $a$ of active features. Details of the generating process are given in Appendix B. We then decompose our synthetic activations using sparse coding and observe how well our metrics predict the true sparsity.

We plot the true sparsity $a$ against each sparsity metric in Fig. 1(a). We see that for metric values less than about 20, the normalized loss and the average coefficient norm closely approximate the true sparsity, while plateauing for greater values. This suggests that these metrics can reliably approximate the correct sparsity level for low sparsities. Meanwhile, number of non-zero features reliably overestimates the true level of sparsity. The final loss is not scale-invariant and so its absolute magnitude cannot be meaningfully compared to the true sparsity.

Second, we test how well our metrics can distinguish between sparse linear data and other datasets. We construct three sparse linear datasets and three non-sparse linear datasets (details in Appendix B), decompose each of the datasets using sparse coding and apply our metrics. The results for average coefficient and normalized loss are shown in Fig. 1(b). We see that the average coefficient norm and normalized loss perform very similarly, and both clearly distinguish between the sparse linear data

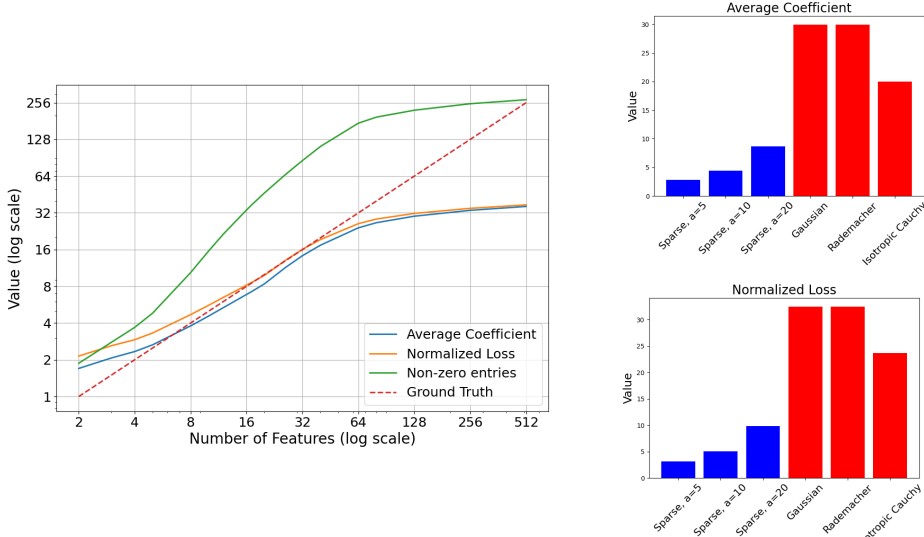

Figure 1: (a) Metric values compared to true sparsity level for synthetic data. (b) Metric values for sparse linear data (blue) compared to non-sparse linear data (red).

and the other datasets. The results for non-zero entries and final loss are shown in Appendix B; both perform less well and so we focus on average coefficient norm and normalized loss from now on.

We also consider ablations of both experiments where we vary the embedding size, the dictionary size, the noise level and the number of ground truth features. We find that our results are robust to changes in the noise level, dictionary size and number of ground truth features, and relatively robust to changes in embedding size, provided the level of sparsity is not too close to the embedding size. For details, see Appendix C.

## 4    DEMONSTRATING SPARSITY IN LANGUAGE MODEL ACTIVATIONS

Now that we have demonstrated that average coefficient norm and normalized loss can reliably distinguish between sparse linear activations and some other classes of distributions, we use them to study sparsity in language model activations. In this section, we focus on the normalized loss, as it is more robust in practice; we present results using average coefficient norm in the appendices.

### 4.1    EMBEDDING LAYERS

First, we use our metrics to assess sparsity in the embedding layers of transformer language models. We pick three classes of models to test on: BERT (Tiny, Mini, Small and Medium) (Turc et al., 2019; Bhargava et al., 2021), TinyStories (1M, 3M and 33M) (Eldan & Li, 2023), and GPT-Neo/GPT-2 (Black et al., 2021; Radford et al., 2019). We use the token embeddings as our set of activations $X$, apply sparse coding, and measure the sparsity of the resulting decomposition using normalized loss. Further experimental details and results for average coefficient are provided in Appendix D.

The normalized loss values we obtain are displayed in Fig. 2, where we have also plotted the sparsity value achieved for a standard Gaussian distribution for reference. We observe that for all models the normalized loss of the token embeddings is much lower than that for a Gaussian, as we would expect if the linearity and sparsity hypotheses held for the embeddings.

Note also that if we compare the normalized loss values in Fig. 2 to the results in Section 3.1 and Appendix C, we see they are within the range where normalized loss correctly predicts ground-truth sparsity on synthetic sparse linear data. This suggests that the normalized loss values in Fig. 2 are a good approximation to the true sparsity level, assuming the linearity and sparsity hypotheses.

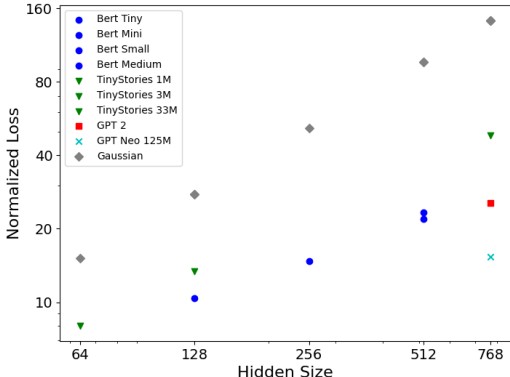

Figure 2: Embedding size versus normalized loss for token embeddings of three different classes of language models.

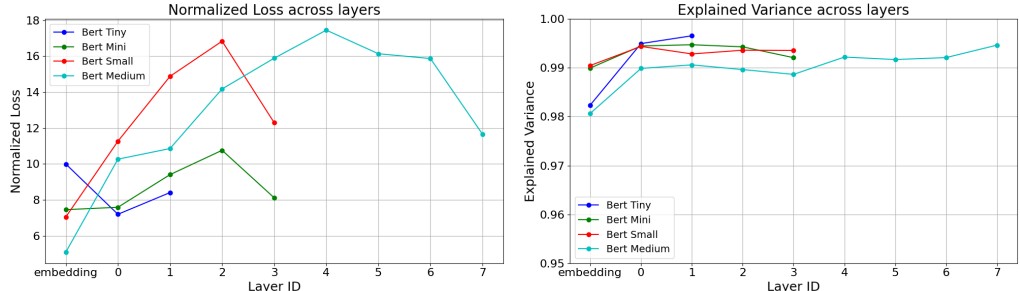

Figure 3: (a) Sparsity of activations by layer for four BERT models. (b) Percentage of activation variance explained by our sparse decomposition for each layer and model.

Second, we observe that the number of active features increases as the embedding size increases, but more slowly. This matches the intuition that the model can disentangle more semantic meanings in a higher dimensional space. We also observe that the BERT architecture leads to slightly more sparse representations than the TinyStories architecture.

As observed in previous work, the feature decompositions produced by our sparse coding method do frequently correspond to natural human interpretations. In Appendix E, we provide examples where our feature decomposition splits tokens into multiple semantically meaningful components, and indicate that the feature directions that we find may be more interpretable than arbitrary directions in embedding space, corroborating the findings of Yun et al. (2021) and Cunningham et al. (2023).

## 4.2 LATER LAYERS

Next, we explore how the level of sparsity changes across layers of a language model. We apply our sparse coding method to all layers of four BERT models (Tiny, Mini, Small and Medium), with activations generated by running the models on Wikipedia abstracts, and plot the resulting normalized loss metric. Full experimental details, along with equivalent results using the average coefficient metric are provided in Appendix F.

The results are plotted in Fig. 3. We find that all layers are sparse according to the normalized loss. In addition, there is a general trend that the embedding layer is the most sparse, with sparsity first decreasing as layer depth increases, before becoming more sparse again in the latest layers. This is consistent with the intuition that the model may initially be detecting a larger number of higher-level features in deeper layers, leading to a reduction in sparsity as layer index increases, before having to extract just the key features for the next token prediction, leading to a subsequent decrease in features stored in the final layers and corresponding increase in the sparsity level.

## 5 RELATED WORKS

**Linear representations**   There is considerable evidence that neural networks learn representations with linear structure: for example, linear operations on Word2Vec embeddings capture semantic meaning (Mikolov et al., 2013; Pham Van et al., 2020), and linear interpolation in the latent space of GANs and VAEs can combine the features of multiple datapoints (Bojanowski et al., 2018; Berthelot et al., 2019). In language models, several works have demonstrated success using linear techniques for locating locating information within models (Meng et al., 2022; Burns et al., 2023) and editing model behaviors (Ilharco et al., 2022; Ravfogel et al., 2022; Turner et al., 2023), as well as observing linear representations in reverse-engineered circuits (Nanda et al., 2023). Such observations motivated the introduction of the linear representation hypothesis in Elhage et al. (2022).

**Feature extraction**   Methods proposed for discovering the features used by neural networks include feature visualization (Erhan et al., 2009; Olah et al., 2017; 2020), saliency maps (Simonyan et al., 2014), feature importance scores (Lundberg et al., 2017), and layer-wise relevance propagation (Bach et al., 2015). For transformer language models, common techniques include visualizing attention patterns (Wang et al., 2023; Bills et al., 2023), gradient-based contrastive explanations (Yin & Neubig, 2022), and more recently methods based on sparse coding (see below).

**Superposition and polysemanticity**   The idea that text embeddings can be modelled as a linear superposition of sparse feature vectors was developed by Faruqui et al. (2015) and Arora et al. (2018). Olah et al. (2020) introduced the notion of polysemanticity in the context of vision models, and hypothesised that polysemantic neurons arise to allow networks to represent more features with a fixed number of neurons. The first theoretical models for polysemanticity and superposition were presented by Elhage et al. (2022) and Scherlis et al. (2022).

**Sparse coding**   The sparse coding problem was introduced by Olshausen & Field (1996; 1997). A variety of methods for learning the sparse dictionary have been proposed, including the method of optimal directions (MOD) (Engan et al., 1999), $k$-SVD (Aharon et al., 2006), and methods using Lagrange duality (Lee et al., 2006). Several works have applied sparse coding to language models using techniques based on autoencoders (Sharkey et al., 2022; Cunningham et al., 2023), or FISTA (Beck & Teboulle, 2009; Yun et al., 2021). The method we use in this paper is a variant of the iterative optimization method used by Yun et al. (2021).

**Automated interpretability**   Recently proposed methods for automating the interpretation of neural networks include using multimodal models to automatically propose interpretations for neurons based on activation patters (Oikarinen & Weng, 2023; Bills et al., 2023), using causal scrubbing to automatically detect circuits within language models (Chan et al., 2022; Conmy et al., 2023), and using singular value decompositions of weight matrices (Millidge & Black, 2022). The previous works most closely related to our current contributions are Yun et al. (2021) and Cunningham et al. (2023), who both use sparse coding to automatically identify a set of feature directions in activation space. They both measure the success of their methods by inspecting sets of examples where a particular feature is active and aiming to identify the common feature between the examples, using either a human labeller or a language model.

## 6 CONCLUSION

In this work, we have developed novel metrics for measuring the success of sparse coding. We have demonstrated that our models can accurately predict the level of sparsity for synthetic models of sparse linear activations introduced by previous works on polysemanticity and superposition. We applied these metrics to measure sparsity in language model activations, finding that activations across various model architectures and all layers were more sparse than several control datasets, providing evidence for the linearity and sparsity hypotheses. Our metrics also allowed us to give quantitative estimates for the number of underlying features, assuming linearity and sparsity, and to show that language model activation sparsity is greatest in the first and final layers of a model, while intermediate layers are relatively less sparse. We hope that through having better quantitative ways to assess feature sparsity and superposition within language models, we can encourage further more detailed and precise studies of this phenomenon, including more rigorous tests of the linearity and sparsity hypotheses, further investigations into the factors that affect the degree of feature superposition within models, and improved methods to disentangle superposed features.

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

## A    DETAILS OF SPARSE CODING ALGORITHM

The algorithm used in this paper for minimizing the sparse coding objective $\mathcal{L}(\Phi; \alpha)$ from (1) over $\alpha \in \mathbb{R}^{m \times n}$ and $\Phi \in \mathbb{R}^{d \times m}$ is as follows. It is inspired by the iterative optimization procedure used by Yun et al. (2021).

We alternate between the following two steps. First, we minimize the objective $\mathcal{L}(\Phi; \alpha)$ over all $\Phi \in \mathbb{R}^{d \times m}$ using several steps of stochastic gradient descent. Second, we update $\alpha$ using a greedy procedure. For each column $\mathbf{x}^{(j)}$ of $X$, we find the current feature vector $\mathbf{f}_i$ that has the largest dot product with $\mathbf{x}$, say $\mathbf{f}_{i_1}$, and set $\alpha_{i_1,j} = \mathbf{x} \cdot \mathbf{f}_{i_1} - \frac{1}{2}\lambda$ (this choice corresponds to projection under the $L^1$ norm). If $\alpha_{i_1,j} > 0$, then we subtract $\alpha_{i_1,j} \mathbf{f}_{i_1}$ from $\mathbf{x}^{(j)}$, find the feature vector with the next largest dot product and repeat. We do this until $\alpha_{i_k,j} \leq 0$ at some step, at which point we set all remaining $\alpha_{s,j} = 0$. We repeat this for each column of $X$ to find the new matrix $\alpha \in \mathbb{R}^{m \times n}$.

We alternate these two steps, updating $\Phi$ and $\alpha$ sequentially until the process converges. In the case where the activations are drawn from a distribution rather than given by a fixed matrix $X$, we resample the activations at each iteration.

We find empirically that the best choice of $\lambda$ is approximately 10% of the maximum activation, i.e. $\lambda \approx 0.1\|\alpha\|_\infty$. Where it is computationally feasible, we use an adaptive scheme to set $\lambda$, first guessing a reasonable choice of $\lambda$, then running our sparse coding procedure once to get a feature coefficient matrix $\alpha$, then updating $\lambda = 0.1\|\alpha\|_\infty$ and iterating until convergence. In practice, we find that we typically only need a couple of steps to converge.

## B    FURTHER DETAILS ON METRIC VERIFICATION EXPERIMENTS

To generate synthetic sparse linear data with $a$ features active on average, we do the following. First, we generate $m = 4d$ feature vectors $\mathbf{f}_1, \ldots, \mathbf{f}_m \in \mathbb{R}^d$ sampled uniformly from the unit sphere. Then, to generate each activation we sample a vector $\alpha$ of feature coefficients by taking each coefficient to be uniform between 0 and 1 with probability $a/4d$ and zero otherwise. We define the proto-activation to be $\hat{\mathbf{x}} = \sum_i \alpha_i \mathbf{f}_i$, so that on average each proto-activation is the sum of $a$ activated vectors. We center the set of proto-activations and finally add Gaussian noise of variance $a\sigma^2/d$ to each proto-activation to get our synthetic activations.

Note that since each active feature is weighted by a factor distributed uniformly between 0 and 1, the expected weighted number of active features will be $a/2$. Hence, we consider the "correct" value of our metric on this dataset to be $a/2$ (rather than $a$).

In our initial experiments in Section 3.1 we take $d = 256$, $\sigma = 0.1$ and use 16384 datapoints and a dictionary size of $8d$. We consider the effects of varying the embedding size, noise level, number of ground-truth features and dictionary size in Appendix C.

For our three non-sparse linear datasets, we use (i) a Gaussian distribution with identity covariance (not sparse), (ii) an heavy-tailed isotropic distribution constructed by sampling from an isotropic Gaussian and then scaling $\|\mathbf{x}\|_2$ to be Cauchy distributed (heavy-tailed but not sparse), and (iii) a $d$-dimensional Rademacher distribution (sparse but not well-represented by a sparse linear combination of features). Each distribution is scaled to have identity covariance.

Fig. 4 shows the results of attempting to use non-zero entries and final loss to distinguish between sparse linear data and other datasets. We see that non-zero entries fails to reliably distinguish the heavy-tailed but non-sparse data, while the final loss fails to reliably distinguish the Gaussian data from the sparse linear data. In order to make the final loss values comparable across datasets, we scale all datasets to have mean 0 and the average $L_2$ norm equal 1.

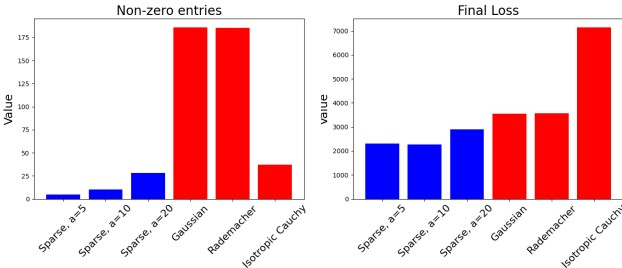

Figure 4: Metric values for sparse linear data (blue) compared to non-sparse linear data (red), with non-zero entries and final loss metrics.

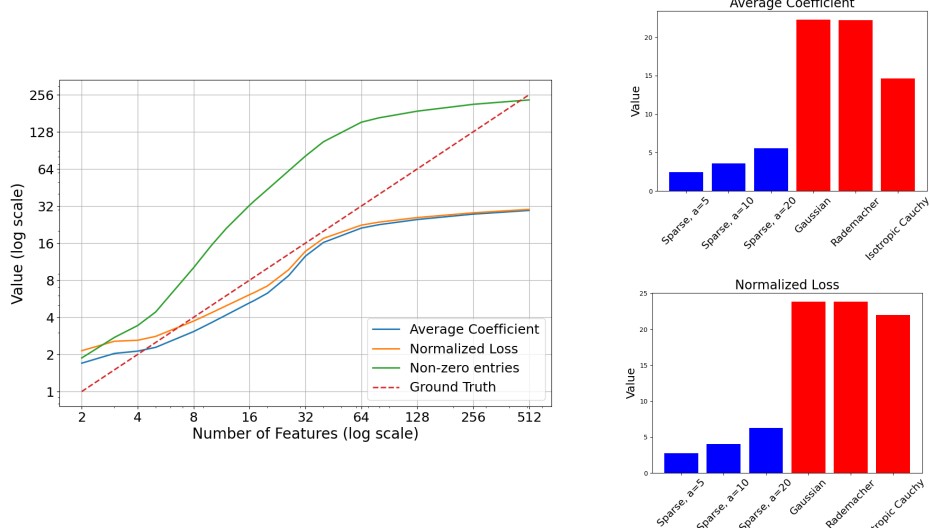

Figure 5: Results of experiment from Section 3.1 with dictionary size of $16d$. (a) Metric values compared to true sparsity level for synthetic data. (b) Metric values for sparse linear data (blue) compared to non-sparse linear data (red).

## C ABLATIONS FOR METRIC VERIFICATION EXPERIMENTS

In this section we assess the robustness of the results from Section 3.1 to changes in the parameters of the problem definition. We consider changing the level of noise in the construction of our synthetic dataset $\sigma$, the size of the embedding dimension $d$, the dictionary size, and the number of ground-truth features.

First, we assess the effect of changing the dictionary size. We keep the same embedding dimension $d = 256$, the same number of ground-truth features $4d$ and the same noise level $\sigma = 0.1$ as in the main text, but consider increasing the dictionary size to $16d$. We plot the results of this experiment in Fig. 5. We see that the results are very similar to Fig. 1; the average coefficient and normalized loss metrics continue to distinguish well between the sparse linear data and the other datasets, and both metrics track the ground-truth sparsity well for metric values below approximately 16. We conclude that our methods are relatively robust to dictionary sizes that are misspecified up to a factor of at least 4.

Second, we assess the effect that changing the noise level $\sigma$ has. We keep the same embedding dimension $d = 256$, the same dictionary size of $8d$ and the same number of ground-truth features at $4d$ as in the main text, but consider decreasing $\sigma$ to $0.05$ or increasing it to $0.2$. The results are plotted in Fig. 6. We see that both the average coefficient and the normalized loss metric continue

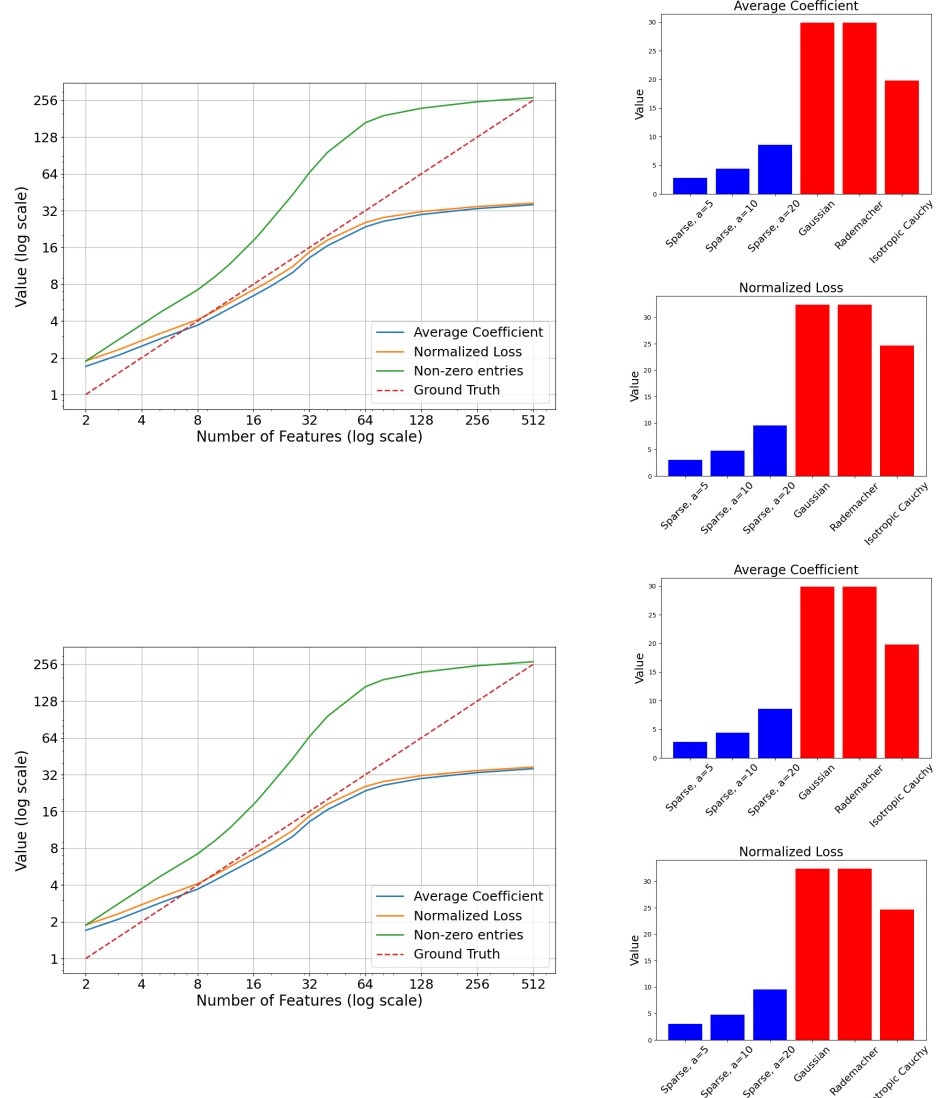

Figure 6: Results of experiment from Section 3.1 with different noise levels: $\sigma = 0.05$ (top) and $\sigma = 0.2$ (bottom). (Left) Metric values compared to true sparsity level for synthetic data. (Right) Metric values for sparse linear data (blue) compared to non-sparse linear data (red).

to distinguish well between sparse linear and other data, and both metrics track the ground-truth sparsity well up to a metric value of about 20.

Third, we assess the effect of changing the embedding size. We fix the dictionary size, number of ground-truth features and noise level as in Section 3.1 but consider taking the embedding size to be either 64 or 512. The results are shown in Fig. 7. We continue to achieve good results for larger embedding sizes. For smaller embedding sizes, our metrics are less able to separate data with $a = 20$ and non-sparse data; this is likely because 20 features is a considerable fraction of the total embedding size when $d = 64$, and so data with $a = 20$ is approaching no longer being appreciably sparse. We see that our metrics still separate linear sparse data with $a = 5, 10$ and the other data sets reasonable well.

Finally, we consider changing the number of ground-truth features from $4d$ to $8d$ (while keeping all other parameters of the synthetic data the same). The results of the experiments from Section 3.1

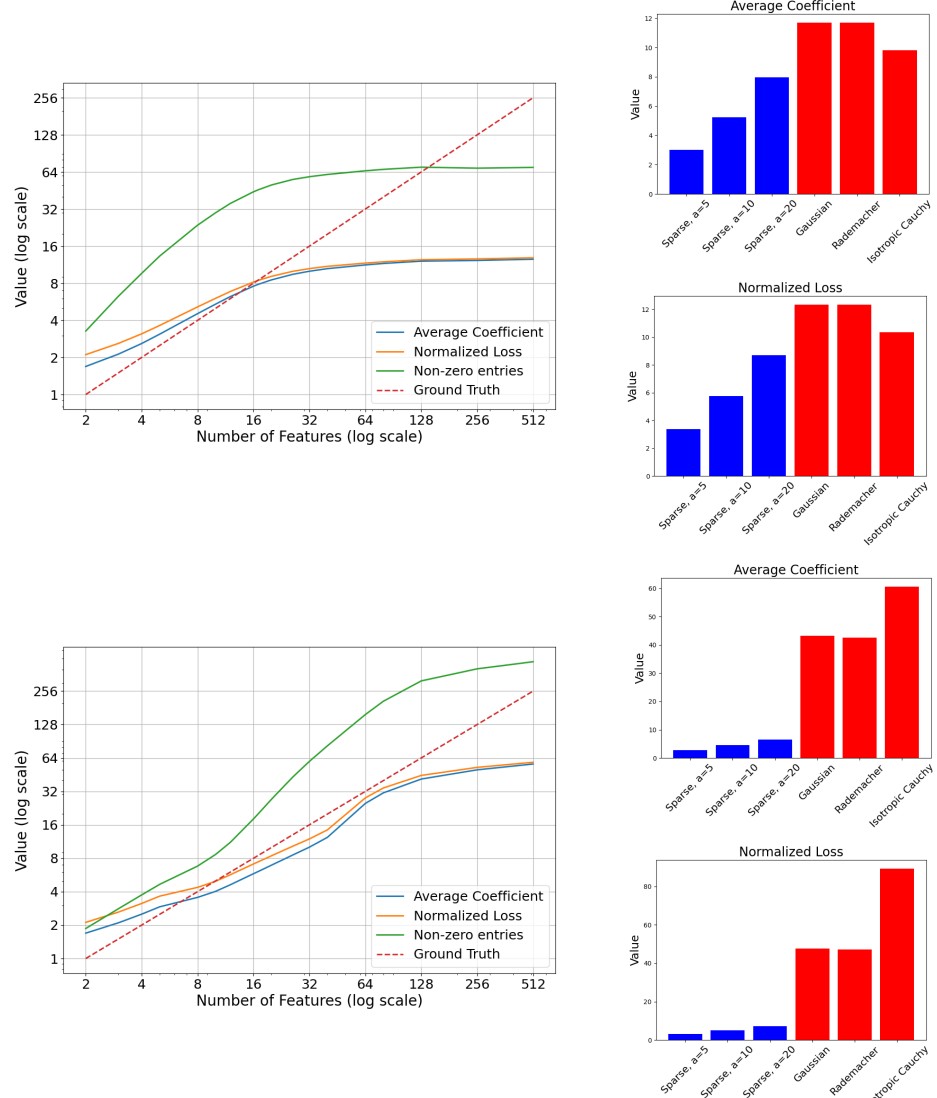

Figure 7: Results of experiment from Section 3.1 with different embedding sizes: 64 (top) and 512 (bottom). (Left) Metric values compared to true sparsity level for synthetic data. (Right) Metric values for sparse linear data (blue) compared to non-sparse linear data (red).

are displayed in Fig. 8. We find that our metrics still accurately track the ground-truth sparsity for metric values up to about 32, and can distinguish between the sparse linear and other datasets.

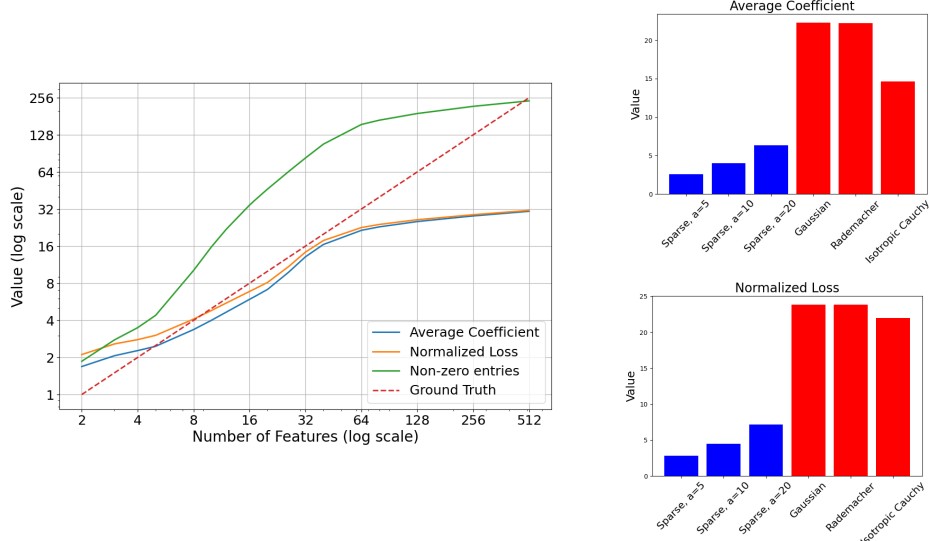

Figure 8: Results of experiment from Section 3.1 with $8d$ ground-truth features. (a) Metric values compared to true sparsity level for synthetic data. (b) Metric values for sparse linear data (blue) compared to non-sparse linear data (red).

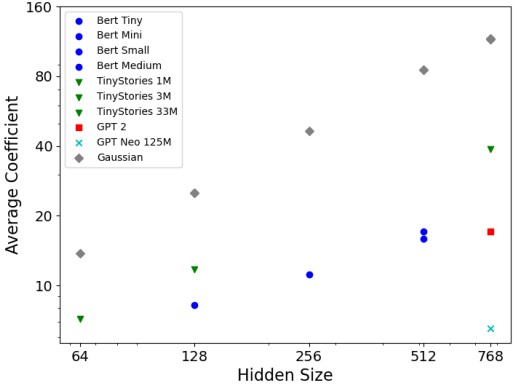

Figure 9: Embedding size versus average coefficient for token embeddings of three different classes of language models.

## D    FURTHER DETAILS ON LANGUAGE MODEL EMBEDDING EXPERIMENTS

For the experiments in Section 4 we use a dictionary size of $8d$. We also center the embedding vectors before applying our sparse coding algorithm. After applying our sparse coding algorithm, the decomposition that we find is able to explain over $90\%$ of the variance in the activations across all models.

In Fig. 9 we show the results of the experiment of Section 4.1 but using the average coefficient metric rather than normalized loss. We find very similar results.

## E    EXAMPLES OF INTERPRETABLE FEATURES IN EMBEDDING SPACE

We provide some initial evidence that the feature directions we find via sparse coding do correspond to human-interpretable features, corroborating the findings of Yun et al. (2021); Cunningham et al. (2023). Given the feature vectors we found in our sparse decomposition of the embeddings of GPT-

| Feature | Max Activating examples | Interpretation |
|---------|-------------------------|----------------|
| Feature 1 | episode, Episode, episode, Episode, episodes, Season, isode, isodes, Season, podcast, odcast, podcast, Podcast, season, flashbacks, Seasons, Ep, Ep, epis, season | TV/Radio shows |
| Feature 2 | Winter, winter, Summer, summer, winter, Winter, Summer, autumn, Autumn, Spring, Spring, summers, winters, spring, Fall, spring, Fall, offseason, seasonal, Halloween | Yearly seasons |
| Feature 3 | year, Year, YEAR, year, Year, decade, month, years, Years, Month, Years, month, years, yr, Month, Months, months, week, season, months | Lengths of time |

Table 1: Max activating examples for top 3 features in the sparse decomposition of `season`.

| Feature | Max Activating examples | Interpretation |
|---------|-------------------------|----------------|
| Feature 1 | Physics, physics, ysics, physic, Math, Math, Chem, math, Chem, particle, chem, chemistry, asm, physicists, math, particles, asms, Chemistry, maths, Phys | Physical sciences |
| Feature 2 | biology, psychology, economics, anthropology, neuroscience, sociology, physiology, biology, astronomy, Biology, chemistry, theology, physics, iology, Economics, Chemistry, Anthropology, ecology, mathematics, ochemistry | Fields of study |
| Feature 3 | interstellar, galactic, galaxies, Interstellar, galaxy, asteroid, Centauri, asteroids, Galactic, astroph, Planetary, Astron, planetary, astronomers, planets, cosmic, spaceship, stellar, Nebula, astronomer | Astronomy |

Table 2: Max activating examples for top 3 features in the sparse decomposition of `physics`.

| Token | Closest embeddings |
|-------|--------------------|
| season | season, Season, season, Season, seasons, offseason, preseason, summer, episode, postseason, episodes, Episode, winter, autumn, playoff, Summer, seasonal, podcast, Winter, Ý, Year, subur, Nitrome, StreamerBot, ÃĥÃ̂ÃĥÃ̂ÃĥÃ̂ÃĥÃ̂ÃĥÃ̂ÃĥÃ̂ÃĥÃ̂ÃĥÃ̂ÃĥÃ̂Ãĥ..., ÃĥÃ̂ÃĥÃ̂ÃĥÃ̂ÃĥÃ̂ÃĥÃ̂, externalTo, decade, episode, Seasons |

Table 3: Closest 30 words to `season` in embedding space

2, we pick a couple of tokens with clear meanings – `season` and `physics` – decompose them into their constituent features, pick the top three features for each with the maximum feature coefficient, and then plot the 20 tokens that maximally activate that feature.

The results are shown in Table 1 and Table 2. We see that for each feature in the decomposition, the maximally activating examples suggest a natural interpretation of that feature, as listed in the third column. For comparison, we also display the 30 words in embedding space which are closest to the embedding direction of `season` in Table 3. We note that this direction is much more polysemantic

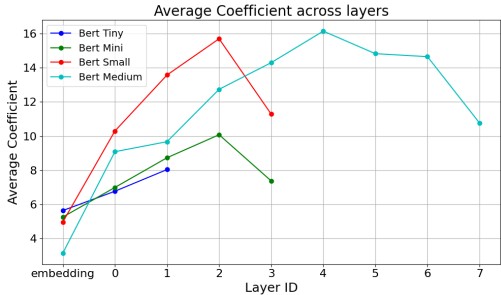

Figure 10: Sparsity of activations by layer for four BERT models, measured using average coefficient norm.

than the rows in Table 1, suggesting that our sparse coding method is succeeding at disentangling superposed meanings.

## F    FURTHER DETAILS OF EXPERIMENTS ON LATER LAYERS IN LANGUAGE MODELS

To form our activation distribution, we sampled model inputs from a dataset of $800$k sentences from Wikipedia abstracts. For each model and layer, we use a dictionary size of $16d$. We fix $\lambda = 0.1$ during training, and then for decomposing into sparse features we use $\lambda = 0.0027, 0.0037, 0.0047, 0.01$ for BERT Tiny, Mini, Small and Medium respectively. This allows us to ensure that we explain at least $98\%$ of the variance for all models and layers, while avoiding needing an adaptive choice of $\lambda$ (for which we did not have sufficient computational resources). We used a batch size of $512$ sentences, and we centered the activations for each batch.

In Fig. 10, we show the results of the experiment of Section 4.2 but using the average coefficient metric rather than the normalized loss. The results are qualitatively similar.

