# OpenReview forum: "Measuring Feature Sparsity in Language Models"
_ICLR.cc/2024/Conference — Submitted to ICLR 2024_

### Official Review · Reviewer_CS4c · 2023-10-29

**Soundness:** 2 fair
**Presentation:** 3 good
**Contribution:** 2 fair
**Rating:** 3
**Confidence:** 3

**Summary:**

The paper explores the hypothesis that the activations of intermediate layers in language models are well-modeled by sparse linear combinations of a set of dictionary elements. Two novel metrics, the Average coefficient norm, and Normalized loss, were presented to quantify the sparsity of embedding layers.

The activation distributions of three classes of transformer language models were analyzed. It was discovered that all layers are sparse and that embedding layers are more sparse in the first and last layers.

**Strengths:**

The paper is well-constructed, with clarity in its presentation and its structure is logical, with each section neatly organized and flowing seamlessly into the next. Its mathematical content is detailed andned, particularly in the background section rigorously defi. The results and data are conveyed clearly, ensuring comprehensibility. The motivation behind the introduced metrics is well-justified, and the conducted experiments are systematically detailed.

**Weaknesses:**

The paper's introduction of two novel sparsity metrics is undermined by inadequate justification for their necessity and a deviation from established interpretations of "ground truth" in sparse activity, challenging the validity of the results. Additionally, the study's reliance on a limited range of datasets and a lack of depth in analysis, particularly in the comparison of large language models' embedding layers to Gaussian models, restricts the generalizability and meaningfulness of the findings. Expanding dataset diversity and deepening the analysis could significantly enhance the paper's contribution to sparsity metrics.

**Questions:**

The main contribution of the paper revolves around two novel sparsity metrics — Normalized loss and Average coefficient. However, the justification for their effectiveness as sparsity metrics remains inadequate.

A primary concern is the definition of “ground truth” in the appendix. It suggests that since each active feature is weighted by a factor uniformly distributed between 0 and 1, the expected weighted number of active features is a/2, and thus, this is considered the “correct” value for the metric. This approach conflicts with the standard definition of sparse activity in datasets and seems more akin to an average coefficient value. Moreover, it contradicts the criteria for a good sparsity metric stated in Section 3, which emphasizes that a meaningful sparsity metric should remain invariant under uniform scaling of all activations. While the proposed metrics themselves are invariant, scaling the “correct” value by 1/2 due to coefficient distribution contradicts this criterion.

If the “Ground Truth” line were adjusted to a, the comparative analysis would differ significantly, potentially showing the Average Coefficient metric as closer to the Ground Truth than the newly proposed metrics. For clarity, this unconventional definition of Ground Truth should be highlighted in the figure caption or the main body, rather than relegated to the appendix.

Additionally, for a novel metric, a broader range of datasets for benchmarking would have been beneficial to demonstrate its applicability.

Furthermore, the conclusion drawn about the higher sparsity of LLMs embedding layers compared to Gaussian models is limited in its meaningfulness. The study could have been strengthened by incorporating a more diverse set of benchmarks and control comparisons, such as analyzing results across various corpora beyond Wikipedia abstracts, to provide a comprehensive evaluation of the proposed metrics.

---

### Official Review · Reviewer_de3y · 2023-10-30

**Soundness:** 1 poor
**Presentation:** 3 good
**Contribution:** 2 fair
**Rating:** 3
**Confidence:** 4

**Summary:**

This paper tests several metrics for underlying feature sparsity, using dictionary learned features from sparse coding. They test these metrics in a synthetic environment and then show that they suggest language models (BERT and TinyStories) have sparser features than might be expected.

**Strengths:**

The topic is extremely interesting, because the sparsity assumption is common in the current discourse around polysemantic neurons. If we can evaluate that assumption, I believe that it would really validate the dictionary learning approach to interpretability, whereas if we can't validate the assumption, our trust in these approaches is limited. I fully believe the hypothesis that LMs involve sparse activation features myself, though I don't have empirical evidence for that.

**Weaknesses:**

My primary issue with this paper is that I'm not convinced that the metrics actually are accurate or meaningful in highly underparameterized settings. This is important because the under parameterized setting is the motivation for the superposition hypothesis. As far as I could tell, the only results that directly confront this are in fig 7. For the embedding size 64, the metrics are highly inaccurate after 32 and for embedding size 512, accuracy falls off before 128. To me, this suggests that the metrics are inaccurate for the main settings we are interested in: under parameterized settings with more distributed features than parameters.

A related issue is that some of these features are likely to be correlated while others are likely to be uncorrelated in practice. That introduces a huge difference between the synthetic and real models. I'm unconvinced by these synthetic experiments. Therefore, I'm unconvinced that the metrics applied to the natural language setting can be trusted, which unfortunately invalidates the other results in the paper.

Minor/citations:
- There is very little reference to the existing work in sparsity and language models. In general, the interpretability literature review appears to be almost entirely image classification work, prior to 2020. There is a very long history of interpretability in NLP, and much of it involves assumptions of sparsity. I recommend reading https://arxiv.org/abs/1812.08951 for an overview, to better engage with the literature in the area, though it only covers up to 2018.
- One topic of discussion in older interpretability literature is whether sparse representations are actually more interpretable. This is taken as an assumption here, but it has been subject to debate for a long time
https://aclanthology.org/2021.acl-short.17/
https://begab.github.io/interpretability-of-sparse-reps

**Questions:**

In section 4.2, are you trying to describe the information bottleneck hypothesis (https://arxiv.org/abs/1703.00810) here? It's a little confusing, but it seems that is what you're trying to do. Closely related work on BERT: https://lena-voita.github.io/posts/emnlp19_evolution.html

---

### Official Review · Reviewer_EBCo · 2023-11-01

**Soundness:** 2 fair
**Presentation:** 3 good
**Contribution:** 1 poor
**Rating:** 3
**Confidence:** 3

**Summary:**

In this paper, the authors proposed metrics to measure the level of sparsity in language models, and they suggest there were evidence that language model activations can be accurately modeled by sparse linear combinations of features. And the metrics for different layers of the model suggest that model activations appear to be sparsest in the first and final layers, and least sparse in middle layers.

**Strengths:**

Paper is clearly written and the paper is easy to read. Appendix provides many details to reconstruct the setup.

Paper used two metrics as baselines: Non-zero entries and Final loss value. And it did the analysis on synthetic dataset.

The analysis result is interesting that both the first and final layers appear to be the sparsest.

**Weaknesses:**

Experiments are not insufficient to support claim that language model activations can be accurately modeled by sparse linear combinations of features. If the claim here is true, we can replace the activations of language model with pruned activations and still get the same performance when compared with unpruned model. There is no experiment to support this.

**Questions:**

When comparing the different metrics in Fig.1, it's required to eyeball which metric performs the best. I think this is not accurate, and it's difficult to judge which metric is better. For example, I can add a 0.5 factor to the Non-zero entries, and maybe it becomes a better metric?

---

### Official Review · Reviewer_5HjV · 2023-11-02

**Soundness:** 3 good
**Presentation:** 3 good
**Contribution:** 2 fair
**Rating:** 5
**Confidence:** 3

**Summary:**

This paper proposes novel metrics for measuring the success of sparse coding on neural network activations in language models. The authors demonstrate the robustness of their metrics on synthetic activations and provide quantitative evidence that neural network activations can be accurately modeled as sparse linear combinations of feature vectors. They also provide a more thorough analysis of the success of sparse coding as compared to previous works, including studying the relative sparsity of various model types and across layers. The main contribution of this work is to introduce more rigorous and quantitative ways of measuring the success of sparse coding methods applied to language models.

**Strengths:**

Firstly, the paper proposes novel metrics for measuring the success of sparse coding on neural network activations in language models. These metrics are more rigorous and quantitative than previous methods, allowing for a more thorough analysis of the success of sparse coding.

Secondly, the authors demonstrate the robustness of their metrics on synthetic activations, providing evidence of the quality of their approach. They also provide a more thorough analysis of the success of sparse coding as compared to previous works, including studying the relative sparsity of various model types and across layers.

Thirdly, the paper is well-written and clear. The authors provide a detailed background and explanation of their methods, making it easy for readers to understand the significance of their findings.

**Weaknesses:**

One potential weakness is that the paper focuses primarily on synthetic models of sparse linear activations rather than real-world language models. While the authors do apply their metrics to language model activations, it would be valuable to see more extensive experiments to validate the effectiveness of their approach.

Additionally, the authors could consider comparing their approach to other methods for measuring sparsity in language models to provide a more comprehensive analysis of the state of the art.

Another potential weakness is that the paper does not provide a detailed discussion of the limitations of their approach. For example, it is unclear how sensitive their metrics are to variations in the input data or how well their approach would generalize to other types of language models.

**Questions:**

see Weaknesses

---

### Meta-Review · Area_Chair_mZWj · 2023-12-05

**Metareview:**

The paper considers the superposition hypothesis and proposes an approach to measuring feature sparcity in language models. Specifically, the authors consider sparse dictionary learning and propose novel metrics for evaluating the learned features: average coefficient norm and normalized loss. The authors perform synthetic experiments as well as experiments with real models and argue that the proposed metrics are sound, and that feature sparsity is higher in first and last layers.

## Strengths

- The questions of feature sparsity and polysemanticity are extremely important in language model interpretabilty
- Novel metrics for dicitionary learning are very important; currently it is challenging to compare feature attributions, which is necessary to make prrogress
- The authors perform synthetic experiments to support the proposed metrics

## Weaknesses

- The reviewers remained unconvinced by the proposed metrics, and whether they are actually meaningful and useful for measuring progress in dictionary learning
- In particular, it is unclear whether the metrics provide useful signal in the underparameterized setting (reviewer de3y)
- Reviewers pointed out that a broader range of benchmarks needs to be considered to support the proposed metric

**Justification For Why Not Higher Score:**

The reviewers raised multiple concerns, and the authors did not provide a rebuttal. In particular, all reviewers currently vote for rejecting the paper. Generally, while the paper pursues a very important direction, more verification is needed to confirm that the proposed metrics are meaningful and useful for measuring progress in dictionary learning.

**Justification For Why Not Lower Score:**

N/A

---

### Decision · Program_Chairs · 2024-01-16

Reject